# Incidence, Risk Factors and Impact on Clinical Outcomes of Bloodstream Infections in Patients Hospitalised with COVID-19: A Prospective Cohort Study

**DOI:** 10.3390/antibiotics10091031

**Published:** 2021-08-24

**Authors:** Andrea Cona, Alessandro Tavelli, Andrea Renzelli, Benedetta Varisco, Francesca Bai, Daniele Tesoro, Alessandro Za, Caterina Biassoni, Lodovica Battaglioli, Marina Allegrini, Ottavia Viganò, Lidia Gazzola, Teresa Bini, Giulia Carla Marchetti, Antonella d’Arminio Monforte

**Affiliations:** 1Clinic of Infectious Diseases, San Paolo Hospital, ASST Santi Paolo e Carlo, Department of Health Sciences, University of Milan, Via di Rudinì 8, 20142 Milan, Italy; alessandro.tavelli@gmail.com (A.T.); andrearenzelli@hotmail.it (A.R.); benedetta.varisco@unimi.it (B.V.); francesca.bai@unimi.it (F.B.); daniele.tesoro@gmail.com (D.T.); marina.allegrini@unimi.it (M.A.); vigano.ottavia@gmail.com (O.V.); gazzolalidia@gmail.com (L.G.); Teresa.bini@unimi.it (T.B.); giulia.marchetti@unimi.ti (G.C.M.); antonella.darminio@unimi.it (A.d.M.); 2Central Health Care Management, ASST Santi Paolo e Carlo, Via di Rudinì 8, 20142 Milan, Italy; alessandro.za@asst-santipaolocarlo.it; 3Microbiology Unit, ASST Santi Paolo e Carlo, Via di Rudinì 8, 20142 Milan, Italy; caterina.biassoni@asst-santipaolocarlo.it (C.B.); lodovica.battaglioli@asst-santipaolocarlo.it (L.B.)

**Keywords:** COVID-19, bacterial super-infections, bloodstream infections, corticosteroid therapy, antimicrobial stewardship

## Abstract

With the aim of describing the burden and epidemiology of community-acquired/healthcare-associated and hospital-acquired bloodstream infections (CA/HCA-BSIs and HA-BSIs) in patients hospitalised with COVID-19, and evaluating the risk factors for BSIs and their relative impact on mortality, an observational cohort study was performed on patients hospitalised with COVID-19 at San Paolo Hospital in Milan, Italy from 24 February to 30 November 2020. Among 1351 consecutive patients hospitalised with COVID-19, 18 (1.3%) had CA/HCA-BSI and 51 (3.8%) HA-BSI for a total of 82 episodes of BSI. The overall incidence of HA-BSI was 3.3/1000 patient-days (95% CI 2.4–4.2). Patients with HA-BSI had a longer hospital stay compared to CA/HCA-BSI and no-BSI groups (27 (IQR 21–35) vs. 12 (7–29) vs. 9 (5–17) median-days, *p* < 0.001) but a similar in-hospital mortality (31% vs. 33% vs. 25%, *p* = 0.421). BSI was not associated with an increased risk of mortality (CA/HCA-BSI vs. non-BSI aOR 1.27 95% CI 0.41–3.90, *p* = 0.681; HA-BSI vs. non-BSI aOR 1.29 95% CI 0.65–2.54, *p* = 0.463). Upon multivariate analysis, NIMV/CPAP (aOR 2.09, 95% CI 1.06–4.12, *p* = 0.034), IMV (aOR 5.13, 95% CI 2.08–12.65, *p* < 0.001) and corticosteroid treatment (aOR 2.11, 95% CI 1.06–4.19, *p* = 0.032) were confirmed as independent factors associated with HA-BSI. Development of HA-BSI did not significantly affect mortality. Patients treated with corticosteroid therapy had double the risk of developing BSI.

## 1. Introduction

Coronavirus disease 2019 (COVID-19) pneumonia is an interstitial pneumonia caused by the severe acute respiratory syndrome coronavirus 2 (SARS-CoV-2) which might lead to acute respiratory failure and death. As for influenza, bacterial co- and super-infections may complicate the disease course [1,2]. The fear of bacterial superinfections, alongside other factors including the lack of antimicrobial stewardship programs especially suited for a pandemic setting, has led to the overuse of antibiotics, despite several studies having shown the incidence of bacterial infections being relatively low in COVID-19 as compared to other viral diseases [3,4,5,6,7]. While the incidence of bloodstream infections (BSIs) and their impact on patient outcome has been extensively described in the intensive care unit (ICU) setting [8,9,10,11,12], data on the epidemiology of BSIs in non-critical wards are scarce. Moreover, risk factors for the development of BSIs during hospitalization for COVID-19 have not been clearly identified yet.

The aim of the study was to describe the burden and epidemiology of community-acquired/healthcare-associated and hospital-acquired bloodstream infections (CA/HCA and HA-BSIs) in patients admitted to San Paolo Hospital in Milan and hospitalised for COVID-19 pneumonia, to evaluate risk factors for HA-BSI and their impact on mortality.

## 2. Results

In the study period, 1825 adult patients were admitted for COVID-19 to the emergency room of San Paolo hospital for COVID-19; 66 (3.6%) died and 408 (22.4%) were discharged within 24 hours. A total of 1351 (74%) patients were hospitalised for SARS-CoV-2 symptomatic infection and were therefore included in the study.

Overall, 69 patients (5%) had a concomitant or subsequent diagnosis of BSI during hospitalisation. In detail, 18 patients (1.3%) were diagnosed with CA/HCA-BSI and 51 (3.8%) with HA-BSI. A total of 82 episodes of BSI were observed, 18 (22%) CA/HCA-BSI and 64 (78%) HA-BSI. 92 microorganisms isolated from blood cultures, namely coagulase-negative staphylococcus spp. or other skin commensals, were considered as contaminants and excluded from the study (see Appendix A). A list of other microbiological findings in the population is provided in the Appendix A.

### 2.1. Baseline and Clinical Characteristics of Patients

Baseline characteristics of 1351 patients included in the study are described in Table 1. A total of 211 cases (16%) were classified as mild, 519 (39%) as moderate, 581 (42%) as severe, and 40 (3%) as critical. Radiologically confirmed pneumonia was present on admission in almost 80% of patients.

On admission, 18 patients received a concomitant diagnosis of CA/HCA-BSI, with 2 patients having a polymicrobial BSI. Patients with CA/HCA-BSI were older and with a higher Charlson Comorbidity Index as compared to patients without BSI. More patients within the CA/HCA-BSI group as compared to the non-BSI group had recently been hospitalised or were long-term-care facility residents (39% CA/HCA-BSI vs. 16% non-BSI group, *p* = 0.01). Patients with CA/HCA-BSI more frequently presented neutrophilic leucocytosis and higher levels of C-reactive protein on admission, but a lower prevalence of radiologically confirmed pneumonia (61% in CA/HCA-BSI vs. 79% in non-BSI, 0.075) (Table 1). Regarding treatments received for COVID-19, fewer patients in this group were treated with steroids (33% CA/HCA-BSI vs. 46% non-BSI, *p* = 0.3). The majority of patients needed low/high oxygen flow or did not need any respiratory support during hospitalisation (89% in CA/HCA-BSI vs. 60% in non-BSI, *p* = 0.01); only two patients required Continuous Positive Airway Pressure therapy (CPAP) or Non-Invasive Mechanical Ventilation (NIMV) (11% vs. 34%, *p* = 0.04) (Table 2).

A total of 64 episodes of HA-BSI were observed in 51 patients with an incidence rate of 3.3/1000patient-days (95% CI 2.4–4.2). Patients who developed HA-BSI had a more severe COVID-19 disease on admission (severe/critical: 60% in HA-BSI vs. 45% in non-BSI group, *p* = 0.03) and a higher prevalence of radiologically confirmed pneumonia (92% in HA-BSI vs. 79% in non-BSI group, *p* = 0.02) (Table 1). A total of 65% of patients in the HA-BSI group received corticosteroids prior to the development of the BSI episode, while immunomodulators were used in only 6% of patients. More patients in this group needed respiratory support via Invasive Mechanical Ventilation (IMV) (22% vs. 6.5% in non-BSI group, *p* < 0.001), NIMV (20% vs. 6% in non-BSI group, *p* < 0.001), while a similar distribution of CPAP use was observed between the groups (27% vs. 28% in non-BSI group, *p* = 0.92) (Table 2).

### 2.2. Clinical and Microbiological Characteristics of Community-Acquired/Healthcare-Associated Bloodstream Infections

At CA/HCA-BSI onset, half of patients were febrile and 15% experienced hypotension, but none required treatment with vasoactive agents. Laboratory findings at onset of BSI were comparable except for a higher level of PCT in the CA/HCA-BSI group (22.35 ug/L vs. 1.02 ug/L in the non-BSI group; *p* = 0.005). Empirical antibiotic therapy was appropriate in 60% of cases, based on susceptibility tests, while targeted therapy was appropriate in 90% of cases. The most common source of CA/HCA-BSIs was the urinary tract (33%) followed by vascular catheter-related BSIs (28%). An equal distribution was observed between Gram-positive and Gram-negative bacteria. The most frequent causative organisms of CA/HCA-BSIs were coagulase-negative staphylococci, followed by *Klebsiella* spp., *Staphylococcus aureus*, *Escherichia coli* and *Pseudomonas* spp. (Table 3).

### 2.3. Clinical and Microbiological Characteristics of Hospital-Acquired Bloodstream Infections

The median time from hospital admission to onset of HA-BSI was 11 days (IQR 4–16) and almost one-fourth of HA-BSI cases occurred in the ICU. The majority of episodes presented with fever and almost one-third with hypotension; vasoactive agents were needed in 13% of cases. Concerning antibiotic therapy, empirical and targeted antimicrobial therapy was appropriate in 58% and 87% of cases, respectively.

The most common source of HA-BSI was the urinary tract (38%) followed by catheter-related BSIs (20%) and respiratory tract; almost 20% of HA-BSIs were of unknown origin. About half of HA-BSIs were caused by Gram-positive bacteria, mainly *Staphylococcus aureus* and coagulase-negative staphylococci. Among Gram-negative pathogens, instead, *Escherichia coli*, *Enterobacter* spp. and non-fermenting Gram-negative bacilli (*Pseudomonas* spp., *Stenotrophomonas maltophilia*, *Acinetobacter baumannii*) were the most frequent causative agents. Two episodes were caused by fungi (*Candida albicans* and *Candida tropicalis*) (Table 3).

### 2.4. Risk Factors for Hospital-Acquired Bloodstream Infections

At univariate analysis, factors associated with an increased risk of HA-BSI during hospital stay were NIMV/CPAP, IMV and corticosteroid treatment. At multivariate analysis, these variables were confirmed as independent factors associated with the onset of HA-BSI (Table 4).

### 2.5. Clinical Outcomes of Patient with CA/HCA and HA Bloodstream Infections

The median overall length of hospital stay was 10 days (IQR 5–18). Of all patients, 7% (101/1351) were admitted to ICU and one-fourth (341/1351) died during hospitalisation (Table 2).

Patients with CA/HCA-BSI had a similar length of hospital stay as compared to patients without BSI (12.5 days vs. 9 days, *p* = 0.054). Contrarily, patients who developed HA-BSI had a longer hospital stay as compared to patients with CA/HCA-BSI and patients without BSI (27 vs. 12.5 vs. 9 days, *p* < 0.001) and more patients were admitted to ICU (22% vs. 0% vs. 7%; *p* < 0.001). In-hospital mortality did not differ significantly among the three groups (31% vs. 33% vs. 25%, *p* = 0.295). Even after adjusting for confounders (age, sex, Charlson Index, CRP and D-dimer levels on admission, severity of disease at admission, calendar period of admission, immunomodulatory agents), BSI was not associated with an increased risk of mortality (CA/HCA-BSI vs. non-BSI aOR 1.27 95% CI 0.41–3.90, *p* = 0.681; HA-BSI vs. non-BSI aOR 1.29 95% CI 0.65–2.54, *p* = 0.463) (Appendix A).

## 3. Discussion

This retrospective observational study provides a comprehensive description of the clinical characteristics of 83 episodes of bloodstream infections in a cohort of 1351 patients hospitalised with COVID-19 in Milan, Italy. To our knowledge, this is the largest monocentric cohort study focused on bloodstream infections in COVID-19 patients available in literature so far.

We found a prevalence of BSI of 5.1%, while the incidence of HA-BSI was 3.3/1000 patient-days with a prevalence of 3.8%. In literature, most studies describe the incidence of bacterial infections in COVID-19 patients without a specific focus on bloodstream infections [3,4,5,6,7,13]. While the incidence of BSI in COVID-19 patients admitted to ICU is well described [8,10,11,12,14,15], only a small number of authors reports data on BSI outside the critical setting [9,16,17]. Ripa et al. [7] recently reported an incidence rate of secondary BSI of 6.7 per 1000 person-days of follow-up, which is slightly higher than the figure we found, while a lower incidence is described in other studies [5,9].

From a microbiological point of view, our results do not differ significantly from those reported by other authors [5,7,8,9,16,18]. In fact, most BSI episodes in our cohort were caused by CONS, *Staphylococcus aureus* and *Escherichia coli*. Surprisingly, half CA/HCA-BSIs were due to Gram-positive bacteria and in one-fourth of cases the primary focus of BSI was a vascular catheter, contrarily to what commonly observed in community infections. This can be partly explained by the fact that almost 40% of patients with CA/HCA-BSIs were long-term facility residents or had been recently hospitalised. Concerning the prevalence of multi-drug resistant bacteria, in our cohort, 18.5% of pathogens overall isolated from blood samples were MDR. Specifically, 58% of CONS and 18% of *Staphylococcus aureus* were methicillin resistant and 17% of *Enterococcus faecium* were vancomycin-resistant. ESBL production was observed in half of *Enterobacter cloacae* and 20% of *Escherichia coli* isolates.

Patients who later developed HA-BSI more frequently had a radiologically documented pneumonia on admission and a more severe clinical presentation. This finding was explained by the fact that, since a more severe disease implies a higher intensity of care (involving, for example, the use of vascular and urinary devices), this exposes individuals to a higher risk of nosocomial infections. In contrast, patients with CA/HCA-BSI were elderly patients, often residents in healthcare facilities, that were admitted to hospital for their frailty, often independently from severity of disease. This is confirmed by the fact that, while pneumonia was radiologically documented only in 60% of these patients, in-hospital mortality was comparable to the one observed in HA-BSI patients.

Risk factors for the development of HA-BSI were analysed. Notably, patients treated with corticosteroid therapy had double the risk of developing BSI during hospitalisation. In our opinion, the independent association between the use of steroids and the increased risk of BSI is the most important finding of our study. Giacobbe et al. [8] recently described an increased risk of developing ICU-acquired BSI in patients receiving methylprednisolone or methylprednisolone plus tocilizumab. On the contrary, immunomodulatory drugs, including steroids and tocilizumab, did not result to be associated with an increased risk of BSI in a recently published work from a Spanish multicentre cohort [18] Corticosteroid use is known to increase the risk of bacterial and opportunistic infections due to its immunosuppressive effects impairing the host response to pathogens [19,20]. Nevertheless, the favourable impact of steroids on patient outcome in severe COVID-19 cases has been proven by the RECOVERY-trial [21] alongside other studies; as a consequence, corticosteroids were introduced in international guidelines for COVID-19 management. Our finding should not be used to discourage the use of steroids, but rather suggests a judicious and wise use of these molecules in order to avoid unintended complications. Regarding immunomodulatory drugs, although we might suppose that their use increases the risk of bacterial superinfections as observed by Buetti et al. [14], this was not observed in our study probably due to the small number of patients who received these agents.

In our analysis, patients who required intensive respiratory support had an increased risk of developing BSI. A similar finding was recently described by Engsbro et al. [16] and Goyal et al. [17]. In our opinion, this association should not be interpreted as a direct consequence of ventilation itself; ventilation should rather be seen as an approximation of disease severity, requiring a higher intensity of care and therefore exposing patients to a higher risk of hospital-acquired infections. Although in-hospital mortality was comparable in all groups, a longer hospital stay was observed in the HA-BSI group. However, we are not able to demonstrate whether BSI was the cause or consequence of longer hospitalisation. Further studies are needed to elucidate this point.

The major strength of this study is the large cohort size; it is in fact one of the largest studies in literature so far with an exclusive focus on bloodstream infections. However, the study has several limitations to be acknowledged. Firstly, the monocentric and retrospective nature of the study entails some biases. Secondly, the study was conducted in one of the geographical regions that were earlier and more severely affected by the ongoing pandemic. The overwhelmed Italian healthcare system may have led to a decreased focus on infection control measures and antimicrobial stewardship principles leading to an increased rate of secondary infections; thus, data from this study cannot be generalized. Thirdly, patients who were admitted to the emergency department and were either discharged or died within 24 hours from admission were excluded from the study. This was done because the goal of the study was to focus on hospitalised patients receiving inpatient care for a minimum of 24 hours, but also because available data in these two settings were limited due to short hospital stay. This represents a limitation of the study, because we cannot exclude the possibility that those patients, especially critical ones, did not have an ongoing BSI. Fourthly, data on empirical and targeted antimicrobial therapy were not collected in all patients but only in patients with BSI. Data on the use of antibiotics in COVID-19 patients suggest an excessive use of antimicrobial agents [3,4] that may exceed the incidence of proven bacterial infections as recently described by Cultrera et al. [12] and may have an impact on the future circulation of MDR pathogens [22,23,24]. Therefore, further studies are needed in this direction in order to promote antimicrobial stewardship principles, as suggested by Huttner et al. in a recent review [25]. Finally, data on the presence of intravascular or urinary catheters were missing in some patients therefore they were not included in our multivariate analysis.

## 4. Materials and Methods

### 4.1. Design and Study Setting

This retrospective observational cohort study was conducted at San Paolo Hospital, a 426-bed university hospital with 20,000 admissions/year in Milan, Italy. All patients admitted to hospital for symptomatic COVID-19 from 24 February to 30 November 2020 were included in the study. Exclusion criteria: age < 18 years; death or discharge from the emergency room within 24 hours. The main objectives of the study were: (i) to describe the microbiological and clinical characteristics of BSIs; (ii) to assess the incidence and risk factors for HA-BSIs; (iii) to evaluate the impact of HA-BSIs on length of stay and in-hospital mortality.

### 4.2. Definitions

Diagnosis of COVID-19 was performed on the basis of a positive real-time reverse transcription polymerase chain reaction (RT-PCR) for SARS-CoV-2 performed on nasopharyngeal throat swab or lower respiratory tract specimens, alongside suggestive clinical and radiological findings. BSI was defined as bacterial growth from a single blood culture in association with clinical findings suggestive for bacterial infection. In case of coagulase-negative *Staphylococcus* spp. or other skin flora commensals, at least two positive blood cultures for the same bacterial species in symptomatic patients were needed to define BSI [26]. Polymicrobial BSIs, with two isolates grown from the same blood culture, were considered as a single clinical episode. Hospital-acquired bloodstream infections (HA-BSIs) were defined as infections arising at least 48 h after hospital admission. Conversely, community-acquired/healthcare-associated bloodstream infections (CA/HCA-BSIs) were defined as infections acquired in the community/long-term facilities and diagnosed within 48 h from admission [27,28]. Severity of COVID-19 on admission was classified as mild (no radiological or clinical evidence of pneumonia), moderate (radiological evidence of pneumonia and PaO_2_/FiO_2_ > 300 mmHg), severe (radiological evidence of pneumonia and PaO_2_/FiO_2_ 100–300 mmHg) or critical (radiological evidence of pneumonia and PaO_2_/FiO_2_ < 100 mmHg). Severity of COVID-19 during hospitalisation was defined by the highest level of respiratory support required and was classified as: no need for oxygen-therapy; low/high flow supplemental oxygen (with a flow of up to 15 L/min), via nasal cannula, simple face mask, venturi mask or non-rebreather mask; continuous positive airway pressure (CPAP) via a helmet device; non-invasive mechanical ventilation (NIMV), mainly bi-level positive airway pressure (BiPAP) via a facemask; invasive mechanical ventilation (IMV).

### 4.3. Data Collection

Demographics, clinical conditions, and microbiological findings of all patients were collected and entered in a database. Electronic medical records were reviewed to include the following data: age, sex, ethnicity, comorbidities (evaluated according to the age-adjusted Charlson comorbidity index (ACCI) [29]), risk factors for SARS-CoV-2 infection, calendar period of hospital admission, symptoms and signs at presentation and during disease course, laboratory findings, radiological findings, PiO_2_/FiO_2_ ratio on admission, severity of COVID-19 on admission, administered COVID-19 treatments (lopinavir/r or darunavir/c, remdesivir, hydroxychloroquine +/− azithromycin, heparin prophylaxis, corticosteroids, immunomodulatory therapy) the highest grade of respiratory support received, length of hospital stay, ICU admission and in-hospital mortality.

Data on microbiological investigations conducted at onset or during hospital stay were collected; these included blood cultures, respiratory cultures (sputum/bronchoalveolar aspirate/bronchoalveolar lavage), urine cultures, pneumococcal and legionella urinary antigen test, PCR for influenza, serology for atypical pulmonary pathogens, multi-drug resistant bacteria (MDR) colonisation. Causative agents and susceptibility test results were investigated using standard microbiologic procedures (BACT/ALERT VIRTUO BioMerieux, as blood culture detection system, VITEK 2 Biomeriux automated system to perform antibiotic susceptibility, MALDI-TOF-MS Biomeriux for microbial identification, STANDARD F analyzer SD Biosensor used to perform qualitative analysis by detecting Legionella pneumophila and Streptococcus pneumoniae antigens in the urine samples and, GeneXpert a real-time RT-PCR-based assay for the detection and differentiation of influenza A and B viral RNA, LIAISON Diasorin for quantitative serology tests).

### 4.4. Ethical Considerations

The study was approved by the Ethic Committee Area 1, Milan (2020/ST/049 and 2020/ST/049_BIS, 3 November 2020) and was conducted according to the guidelines of the Declaration of Helsinki. All patients gave informed consent for the use of their anonymised data for research purposes.

### 4.5. Statistical Analysis

Categorical variables were analysed using absolute numbers and percentages, while continuous variables were analysed using the median and interquartile range (IQR). Chi-square and Kruskal–Wallis tests were used when appropriate to compare characteristics of patients who had CA/HCA-BSI or HA-BSI and those who did not.

The incidence of BSI and HA-BSI was calculated by univariable Poisson regression with 95% confidence interval and defined as the number of events per 1000 patient-days. Factors associated with development of HA-BSI were analysed using an unadjusted and adjusted logistic regression model. Covariates included in the model were chosen *a priori* based on variables described in literature. Impact of BSI on mortality was evaluated by an unadjusted and adjusted logistic regression model.

A *p*-value < 0.05 was considered as statistically significant. All analyses were performed using Stata (v14, StataCorp, College Station, TX, USA). The manuscript was edited in accordance with the Strobe statement (see Appendix A).

## 5. Conclusions

In conclusion, in our study we found a relatively low incidence of BSI in patients hospitalised with SARS-CoV-2 infection. Development of HA-BSI did not significantly affect mortality but was associated with longer hospital stays. Corticosteroid therapy was independently associated with increased risk of acquiring BSI during hospitalisation. Our findings are aimed at promoting, rather than discouraging, a judicious use of steroids in COVID-19 patients, in order to avoid bacterial superinfections that may complicate the clinical course of a disease in which therapeutic options are still limited.

## Figures and Tables

**Table 1 antibiotics-10-01031-t001:** Demographics and clinical characteristics of 1351 patients hospitalised with COVID-19.

	Patients without BSI N = 1282 (94.9%)	Patients with CA/HCA-BSI N = 18 (1.3%)	Patients with HA-BSI N = 51 (3.8%)	*p* Value	Overall N = 1351
**Gender, Male, N (%)**	797 (62.2%)	10 (55.6%)	35 (68.6%)	0.541	842 (62.3%)
**Age, years, median (IQR)**	68 (54–80)	75 (58–82)	63 (57–80)	0.444	68 (55–80)
**Comorbidities, N (%)**					
Hypertension	619 (48.3%)	12 (66.7%)	25 (49.0%)	0.300	656 (48.6%)
Diabetes	250 (19.5%)	3 (16.7%)	14 (27.5%)	0.356	267 (19.8%)
Cardiovascular diseases	378 (29.5%)	7 (38.9%)	18 (35.3%)	0.471	403 (29.8%)
Cerebrovascular diseases	115 (9.0%)	4 (22.2%)	2 (3.9%)	0.065	121 (9.0%)
COPD/asthma	171 (13.3%)	2 (11.1%)	8 (15.7%)	0.854	181 (13.4%)
Chronic liver diseases	50 (3.9%)	0 (0.0%)	2 (3.9%)	0.694	52 (3.9%)
Solid or haematological malignancy	108 (8.4%)	5 (27.8%)	2 (3.9%)	0.007	115 (8.5%)
Chronic kidney disease	94 (7.3%)	2 (11.1%)	5 (9.8%)	0.676	101 (7.5%)
HIV infection /AIDS	12 (0.9%)	1 (5.6%)	1 (2.0%)	0.126	14 (1.0%)
Rheumatic Diseases	26 (2.0%)	0 (0.0%)	0 (0.0%)	0.490	26 (1.9%)
**Age Unadjusted Charlson score, median (IQR)**	1 (0–2)	2 (0–3)	1 (0–2)	0.077	1 (0–2)
**Calendar period of hospital admission, N (%)**				0.553	
February–July 2020	555 (43.3%)	8 (44.4%)	26 (51.0%)		589 (43.6%)
August–November 2020	727 (56.7%)	10 (55.6%)	25 (49%)		762 (56.4%)
**Risk factors for SARS-CoV-2 infection, N (%)**				0.142	
Close contact/household	122 (9.5%)	1 (5.6%)	8 (15.7%)		131 (9.7%)
Healthcare worker	52 (4.1%)	0 (0.0%)	1 (2%)		53 (3.9%)
Hospitalisation last 30 days/ long-term care facility	203 (15.8%)	7 (38.9%)	11 (21.6%)		221 (16.4%)
Unknown/other	905 (70.6%)	10 (55.5%)	31 (60.7%)		946 (70.0%)
**COVID-19 Severity at admission, N (%)**				0.251	
Mild	203 (15.8%)	4 (22.2%)	4 (7.84%)		211 (15.6%)
Moderate	498 (38.9%)	5 (27.8%)	13 (31.4%)		519 (38.4%)
Severe	542 (42.3%)	9 (50.0%)	30 (58.8%)		581 (43.0%)
Critical	39 (3.0%)	0 (0.0%)	1 (2%)		40 (3.0%)
**Pneumonia at X-ray or CT scan, N (%)**	1007 (78.6%)	11 (61.1%)	47 (92.2%)	0.012	1065 (78.8%)
**Laboratory findings at admission** Haemoglobin/dL, median (IQR)	13.4 (12–14.7)	11.4 (10–13.2)	13.9 (12.3–15.1)	0.008	13.4 (12–14.7)
Platelets 10^3^/uL, median (IQR)	208 (161–264)	186 (110–228)	203.5 (161–284)	0.102	207 (161–263)
Leukocytes count, 10^3^/uL, median (IQR)	6.82 (5.07–9.50)	9.94 (4.97–12.96)	7.1 (5.28–9.84)	0.118	6.85 (5.07–9.54)
Neutrophils, 10^3^/uL, median (IQR)	4.96 (3.41–7.57)	8.42 (3.64–11.60)	5.67 (3.79–8.82)	0.780	5.01 (3.42–7.66)
Lymphocyte count, 10^3^/uL, median (IQR)	1.02 (0.7–1.43)	0.83 (0.41–1.52)	0.98 (0.63–1.34)	0.586	1.02 (0.69–1.43)
CRP, mg/L, median (IQR)	54.7 (23.8–96.3)	65.25 (34.1–110.5)	65.7 (33.5–106.1)	0.174	55.3 (24.4–97.3)
LDH, U/L, median (IQR)	292 (228–389)	237 (205–347)	348 (235–417)	0.142	293 (226–390)
D-Dimer, ng/mL, median (IQR)	355 (214–688)	496.5 (406–1911)	383 (264–1549)	0.051	358 (216–692)

CA/HCA-BSI = community-acquired/healthcare-associated bloodstream infection; HA-BSI = hospital-acquired bloodstream infection; COPD = Chronic obstructive pulmonary disease; HIV = Human immunodeficiency virus; AIDS = acquired immunodeficiency syndrome; CT = Computed Tomography; CRP = C-reactive protein; LDH= Lactate dehydrogenase.

**Table 2 antibiotics-10-01031-t002:** Treatment and clinical outcomes of 1351 patients hospitalised with COVID-19.

	Patients without BSI N = 1282 (94.9%)	Patients with CA/HCA-BSI N = 18 (1.3%)	Patients with HA-BSI N = 51 (3.8%)	*p* Value	Overall N = 1351
**COVID-19 treatment, N (%)**					
lopinavir/r or darunavir/c	130 (10.1%)	0 (0.0%)	7 (13.7%)	0.253	137 (10.1%)
remdesivir	136 (10.6%)	2 (11.1%)	6 (11.8%)	0.964	144 (10.7%)
hydroxychloroquine +/− azithromycin	414 (32.3%)	5 (27.8%)	20 (39.2%)	0.534	439 (32.5%)
heparin prophylaxis	874 (68.2%)	10 (55.6%)	41 (80.4%)	0.091	925 (68.5%)
corticosteroids	584 (45.6%)	6 (33.3%)	33 (64.7%)	0.015	623 (46.1%)
immunomodulators	57 (4.5%)	0 (0.0%)	3 (5.9%)	0.581	60 (4.4%)
**Highest grade of O_2_ therapy, N (%)**				<0.001	
IMV	83 (6.5%)	0 (0.0%)	11 (21.6%)		94 (7.0%)
NIMV	73 (5.7%)	1 (5.6%)	10 (19.6%)		84 (6.2%)
CPAP	360 (28.1%)	1 (5.6%)	14 (27.5%)		375 (27.8%)
O_2_ low/high flow	541 (42.2%)	12 (66.7%)	14 (27.5%)		567 (42.0%)
No O_2_ therapy	225 (17.6%)	4 (22.2%)	2 (3.9%)		231 (17.1%)
**Length of hospital stay, Median days (IQR)**	9 (5–17)	12.5 (7–29)	27 (21–35)	<0.001	10 (5–18)
**ICU admission, N (%)**	90 (7.0%)	0 (0.0%)	11 (21.6%)	<0.001	101 (7.5%)
**Death, N (%)**	319 (24.9%)	6 (33.3%)	16 (31.4%)	0.421	341 (25.2%)

CA/HCA-BSI = community-acquired/healthcare-associated bloodstream infection; HA-BSI = hospital-acquired bloodstream infection; IMV = Invasive Mechanical Ventilation; NIMV = Non-Invasive Mechanical ventilation; CPAP = Continuous positive airway pressure therapy; ICU = Intensive Care Unit.

**Table 3 antibiotics-10-01031-t003:** Clinical and microbiological characteristics of 82 bacteremic episodes.

	Total Episodes of BSI N = 82 (100%)	Episodes of CA/HCA-BSI N = 18 (22%)	Episodes of HA-BSI N = 64 (78%)	*p* Value
**Origin of sepsis**				0.535
Respiratory	6 (7.4%)	0 (0.0%)	6 (9.4%)	
Urinary	30 (36.6%)	6 (33.3%)	24 (37.5%)	
Catheter-related	18 (21.9%)	5 (27.8%)	13 (20.3%)	
Intra-abdominal	11 (13.4%)	4 (22.2%)	7 (10.9%)	
Cutaneous	2 (2.4%)	0 (0.0%)	2 (3.1%)	
Other/unknown	15 (18.3%)	3 (16.7%)	12 (18.8%)	
**Gram**				0.721
Positive	41 (48.8%)	10 (50.0%)	31 (48.4%)	
Negative	41 (48.8%)	10 (50.0%)	31 (48.4%)	
Fungi	2 (2.4%)	0 (0.0%)	2 (3.2%)	
**Causative agents, N (%)**				0.448
*Staphylococcus aureus* ^a^	11 (13.3%)	2 (10%)	9 (14.3%)	
*Coagulase-negative staphylococci* ^b^	12 (14.4%)	5 (25%)	7 (11.1%)	
*Enterococcus faecium* ^c^	6 (7.3%)	1 (5%)	5 (8%)	
*Enterococcus faecalis*	7 (8.4%)	1 (5%)	6 (9.5%)	
*Streptococcus pneumoniae*	1 (1.2%)	0 (0%)	1 (1.6%)	
*Corynebacterium* spp. ^d^	2 (2.4%)	0 (0%)	2 (3.2%)	
*Escherichia coli* ^e^	10 (12%)	2 (10%)	8 (12.7%)	
*Klebsiella* spp. ^f^	7 (8.4%)	3 (15%)	4 (6.3%)	
*Enterobacter cloacae* ^g^	6 (7.3%)	0 (0%)	6 (9.5%)	
*Proteus mirabilis*	1 (1.2%)	1 (5%)	0 (0%)	
*Pseudomonas* spp. ^h^	6 (7.3%)	2 (10%)	4 (6.3%)	
*Acinetobacter baumannii*	1 (1.2%)	0 (0%)	1 (1.6%)	
*Stenotrophomonas maltophilia*	2 (2.4%)	0 (0%)	2 (3.2%)	
*Serratia marcescens*	5 (6%)	1 (5%)	4 (6.3%)	
*Raoultella ornithinolytica*	1 (1.2%)	1 (5%)	0 (0%)	
*Bacterioides fragilis*	1 (1.2%)	0 (0%)	1 (1.6%)	
*Listeria monocytogenes*	1 (1.2%)	0 (0%)	1 (1.6%)	
*Campylobacter jejuni*	1 (1.2%)	0 (0%)	1 (1.6%)	
*Lactobacillus casei*	1 (1.2%)	1 (5%)	0 (0%)	
*Candida* spp. ^i^	2 (2.4%)	0 (0%)	2 (3.2%)	
**Fever at onset of BSI (T°C > 37.5)**	58 (69.9%)	10 (50%)	48 (76.2%)	0.026
**Hypotension at onset of BSI**	20 (24.1%)	3 (15.0%)	17 (27.0%)	0.275
**Laboratory findings at onset, median (IQR)**				
WBC, 10^3^/Ul	11.62 (7.09–15.38)	11.14 (6.78–14.38)	11.96 (7.34–15.82)	0.537
N, 10^3^/Ul	9.82 (6.01–13.65)	8.48 (4.27–13.19)	10.11 (6.63–13.65)	0.489
CRP, mg/L	104.6 (53.8–118.2)	88.9 (30.45–112.9)	105.5 (67.9–119.3)	0.237
PCT, ug/L	1.02 (0.18–5.02)	22.35 (2.59–48.55)	0.79 (0.16–2.94)	0.005
**Onset in ICU, N (%)**	15 (18.1%)	0 (0.0%)	15 (23.8%)	0.016
**Days from admission to BSI, median (IQR)**	13 (8–19)	0 (0–0)	11 (4–16)	<0.001
**Appropriate empiric ATB**	49 (57.8%)	12 (60%)	37 (57.8%)	0.822
**Appropriate targeted ATB** ^j^	74 (88.0%)	18 (90.0%)	56 (87.5%)	0.136
**Vasoactive agents use**	8 (9.6%)	0 (0.0%)	8 (12.7%)	0.094

CA/HCA-BSI = community-acquired/healthcare-associated bloodstream infection; HA-BSI = hospital-acquired bloodstream infection; WBC = White Blood Cells; N = Neutrophils; CRP = C-reactive protein; PCT = Procalcitonin; ICU = Intensive Care Unit; ATB = Antibiotic. ^a^ 2/11 Staphylococcus aureus were methicillin-resistant (18%); ^b^ 7/12 Clinically significant Coagulase-negative staphylococci were methicillin-resistant (58%); ^c^ 1/6 Enterococcus faecium were vancomycin-resistant (17%); ^d^
*C. jeikeium* (1/2), *C. striatum* (1/2); ^e^ 2/10 *E.coli* were third-generation cephalosporin–resistant (20%); ^f^
*K. Pneumoniae* (6/7), *K. Oxytoca* (1/7). 0/7 *Klebsiella* spp. were third-generation cephalosporin–resistant (0%), 1/7 was piperacillin/tazobactam-resistant (14%); ^g^ 3/6 Enterobacter cloacae were third-generation cephalosporin–resistant (50%); ^h^
*P. aeruginosa* (5/6), *P. putida* (1/6); ^i^
*C. albicans* (1/2), *C. tropicalis* (1/2); ^j^ one patient (1 episode, 5% of CA/HCA-BSI and 1, 2% of all BSI) died before the pathogen was typified, so no targeted antibiotic therapy was administered. Two patients had a polymicrobial CA/HCA-BSI.

**Table 4 antibiotics-10-01031-t004:** Uni- and multi-variable analysis of risk factors for the development of hospital acquired BSI during hospitalisation for COVID-19.

	OR	95% CI	*p* Value	AOR *	95% CI	*p* Value
**Age**, per 10 years older	1.03	0.87–1.22	0.713	1.03	0.83–1.29	0.764
**Gender**, male (vs. female)	1.33	0.73–2.43	0.415	1.04	0.55–1.94	0.908
**Charlson age unadjsuted**, per one-point raise index	1.10	0.87–1.39	0.352	1.16	0.88–1.53	0.288
**Max O_2_-tp** (vs. no O_2_-tp or high/low flow O_2_)						
NIMV/C-PAP	2.65	1.39–5.05	0.003	2.09	1.06–4.12	0.034
IMV	6.34	2.84–14.13	<0.001	5.13	2.08–12.65	<0.001
**Calendar Period of Admission**, August–November 2020 (vs. February–July 2020)	0.73	0.41–1.28	0.279	0.65	0.33–1.27	0.213
**Anti-inflammatory treatment**						
Corticosteroids	2.19	1.22–3.93	0.009	2.11	1.06–4.19	0.032
Immunomodulators	1.34	0.41–4.44	0.629	0.96	0.28–3.29	0.946

*Adjusted for all the factors showed in table. OR= Odd Ratio; aOR = adjusted Odd Ratio; IMV = Invasive Mechanical Ventilation; NIMV = Non-Invasive Mechanical ventilation; CPAP= Continuous positive airway pressure therapy.

## Data Availability

Data are available from the corresponding author upon reasonable request.

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
