# Peer review of "Incidence, Risk Factors and Impact on Clinical Outcomes of Bloodstream Infections in Patients Hospitalised with COVID-19: A Prospective Cohort Study"

_antibiotics, 2021, doi:10.3390/antibiotics10091031_

Round 1

Reviewer 1 Report

  1. The aim of the study, according to the last paragraph of the Introduction was “the burden and epidemiology of community-acquired/healthcare-associated and hospital-acquired bloodstream infections (CA/HCA and HA-BSIs)” and “to evaluate risk factors for HA-BSI and impact on mortality”. Please, clarify why not analyze the risk factors and impact on mortality of CA/HCA-BSI. This information me be useful to know which patients would require antimicrobial therapy at admission.
  2. Why did the authors exclude patients in the first 24 h (alive or dead)? Specially, dead patients may have BSI.
  3. Please, may use the WHO ordinal scale of severity, to assess the severity of patients? It can facilitate the translation of the results to other clinical settings.
  4. It´s not clear if the two patients with polymicrobial CA/HCA-BSI had two isolates in the same blood cultures. If it is true, then the number of patients with CA/HCA-BSI is 18, because of it was the same clinical episode. Consequently, Table 3 must be modified.
  5. 45.3% and 50% of patients without BSI and CA/HCA-BSI had severe or critical disease, with mortality rates of 24.9% and 33.3%, respectively. However, only 12.2% and 5.6% received IMV or NIMV. It seems inconsistent. Please, clarify it.
  6. It is stated that “HA-BSI had a longer hospital stay as compared to patients with CA/HCA-BSI and patients 167 without BSI (27 vs. 12.5 vs. 9 days, p<0.001) and more patients were admitted to ICU (22% 168 vs. 0% vs. 7%; p<0.001).” It seems that patients admitted to ICU (the cause) had longer stays and more HA-BSI (the consequences). Please, clarify it.
  7. A sentence with the major conclusions must be included at the end of the Abstract and Discussion.

Reviewer 2 Report

The authors describe the burden and epidemiology of community-acquired/healthcare-associated and hospital-acquired bloodstream infections in patients admitted for COVID-19 pneumo-54, to evaluate risk factors for HA-BSI and impact on mortality. Overall the study and the hypothesis is clear and well described but I think that the authors should underline the novelty of this study and explain what add at the current knowledge about this topic.

I have the following major points that should be addressed by the authors.

Please clarify how do you define the severity of COVID-19 disease. (reported in table 1)

Please clarify the terms used in table 2 “Highest grade of O2 therapy, N (%)” Please give details about the use of “high flow”. What do you mean ? Nasal high flow oxygen therapy? And about the NIV --- which interface and support was used?

I think that the paper is well detailed in particular regarding the therapy. In my opinion, the discussion should be revised taking into account some recent analysis performed on similar topic and compared the results obtained (PMID: 33923992)

Another relevant analysis is the occurrence of resistance. Do you have any data about it? I think that this analysis could be useful and add relevant data in literature.

Overall, the sample size is limited. Please can you try to justify the sample size base on power analysis in order to guarantee that the conclusion are not influenced by limited number of patient population.

Reviewer 3 Report

Commentaries to the authors: I sincerely consider that the topic of the article is really interesting because information about BSI on CPVID-19 patients is scarce in the literature. However, the study has marked biases that make the results difficult to interpret and not generalizable to the rest of the population.

 Major comments:

  1.    In the introduction section, lines 46-48, the authors state that the indiscriminate use of antibiotics in COVID is a consequence of the fact that infections can complicate the course of the disease. I think it is not a causal relationship. Infections and super infections can complicate the course of the disease, but they are not the cause of indiscriminate administration of antibiotics to COVID patients. Other more likely causes are the care of patients by non-specialized physicians and the failure of antibiotic administration programs in a pandemic situation.
  2.  Regarding the methods section, the methods of the study are explained in an extensive and precise way. However, the main problem lies in the fact that the CA /HCA BSI group is very heterogeneous because the authors have mixed patients who are admitted from home, with patients who have been admitted for a long stay in health-care centers, whose complications are similar to nosocomial ones. As an example, the marked catheter-related BSI rate detected in this group. Moreover, considering the low percentage of pneumonia in the group of patients with BSI CA / HCA, it seems plausible that some of these patients were admitted for sepsis and not for COVID symptoms.
  3. As for the results section, is well exposed. However, it should be noted that the N size of BSI groups is small. On the other hand, I believe that the tables should agree with the text of the results section, so instead of exposing the comparative analysis between the 3 groups, the comparative analysis between non-BSI and each of the other groups separately.

Minor comments:

  1.    Some small grammatical and spelling mistakes should be checked, such as not adding articles to the sentences.
  2.  ST1. Can the authors explain why Campylobacter jejunii has been considered a contaminating microorganism?
  3. lines 121-123. the names of the microorganisms must be written in italics.
  4. It would be interesting to add the table for the univariate and multivariate analysis of mortality.
  5. I suggest that the authors review the new published literature on the topic. For example, the article by Abelenda-Alonso G et al. (Clin Microbiol Infect 2021 Jul 6;S1198 743X(21)00372-4.

Round 2

Reviewer 1 Report

Thanks for your answers and clarifications. I understand the limitations of the data from the clinical records, and the small number of patients, especially in the CA/HCA-BSI. However, these limitations affect seriously the conclusions of this work aimed to gain insights on the incidence, the risk factors and impact on clinical outcomes of the BSI in patients hospitalized with COVID-19. Regarding if the longer hospitalization in the HA-BSI cases is secondary to the BSI or the BSI occurred because a longer previous hospitalization, it should be useful to know the days of hospital stay before the BSI episodes. Finally, the inconsistent data between the severity classification used in this work and the patients receiving IMV or NIMV, clarified by the authors, may be solved using the WHO ordinal scale, without needing to explain it. 

Reviewer 3 Report

I agree with the changes made by the authors in the manuscript.

Author Response

Thank you very much for your comments